# Enhanced Photoluminescence of Europium-Doped TiO_2_ Nanoparticles Using a Single-Source Precursor Strategy

**DOI:** 10.3390/molecules29245824

**Published:** 2024-12-10

**Authors:** Violaine Mendez, Marlène Fabre, Thibaut Cornier, Françoise Bosselet, Stéphane Loridant, Sarah Asaad, Stéphane Daniele

**Affiliations:** 1IRCELYON, CNRS, University Claude Bernard Lyon 1, UMR 5256, F-69100 Villeurbanne, France; violaine.mendez@gmail.com (V.M.); marlene.fabre@ircelyon.univ-lyon1.fr (M.F.); thibaut.cornier@ircelyon.univ-lyon1.fr (T.C.); francoise.bosselet@ircelyon.univ-lyon1.fr (F.B.); stephane.loridant@ircelyon.univ-lyon1.fr (S.L.); 2CP2M-ESCPE Lyon, CNRS, University Claude Bernard Lyon 1, UMR 5128, 43 Bd du 11 Nov. 1918, CEDEX, 69616 Villeurbanne, France; sara.asad@univ-lyon1.fr

**Keywords:** heterometallic alkoxides, single-source precursor, sol–gel process, titania, Eu^3+^ doping, luminescent properties

## Abstract

TiO_2_:Eu^3+^ nanoparticles with varying europium concentrations were successfully synthesized via a one-pot sol–gel approach using a molecular heterometallic single-source precursor (SSP) Eu-Ti. For comparison, nanomaterials with similar europium levels were also produced by impregnating europium salts onto the same TiO_2_ substrate. All the nanomaterials were thoroughly characterized using Eu elemental analysis, powder X-ray diffraction (XRD), scanning (SEM), transmission (TEM), scanning transmission electron microscopy (STEM), Brunauer–Emmett–Teller (BET) analysis, thermogravimetric analysis (TGA), X-ray photoelectron spectroscopy (XPS), Raman spectroscopy, and photoluminescence (PL). This low-temperature synthesis yielded crystalline powders, and calcination at 400 °C was performed to remove surface organic impurities, enabling a precise comparison of the final nanomaterials. While both preparation methods produced materials with similarly dispersed and localized dopants on the TiO_2_ surface, photoluminescence studies revealed that the SSP-derived nanomaterials exhibited significantly superior electro-optical properties. This enhanced efficiency is attributed to the co-hydrolysis of both reactants, which facilitates an optimized interface between the crystalline TiO_2_ core and the dopant-rich amorphous surface, thereby enabling far more effective charge transfer than that achieved by impregnation.

## 1. Introduction

Rare earth-doped semiconductor nanomaterials have become widely utilized in optoelectronic devices [1]. Among semiconductors, TiO_2_ stands out as a host material in rare earth (RE) photoluminescent (PL) technologies, thanks to its low synthesis cost, mechanical durability, and chemical stability. Europium (Eu^3+^)-activated phosphors, recognized for their high fluorescence efficiency, have recently emerged as promising red phosphors for white LEDs [2] and biological tracking applications [3]. High PL performance is essential in these fields, as it influences not only manufacturing and recycling costs but also the amount of nanomaterial required to meet safety thresholds for toxicity. PL efficiency is largely determined by the synthesis method, which impacts the distribution of Eu^3+^ ions, their local environment, and TiO_2_ crystallinity. Due to the significant size difference between Ti^4+^ (0.605 nm) and Eu^3+^ (0.947 nm) ions in octahedral coordination [4], the direct incorporation of RE ions into the TiO_2_ lattice is challenging. This often results in Eu^3+^ ions being distributed on the (sub-)surface of TiO_2_ nanocrystals, where their proximity can reduce luminescence through cross-relaxation or Eu_2_O_3_ segregation. Furthermore, crystal defects and moderate- to high-temperature annealing (500–1000 °C) can weaken emission intensity, either by expelling Eu^3+^ ions from the TiO_2_ matrix or by forming the highly symmetric, low-PL Eu_2_Ti_2_O_7_ phase [5]. Thus, achieving a high dispersion of Eu^3+^ ions at meaningful concentrations within TiO_2_ nanoparticles is essential for optimizing their performance in optoelectronic applications.

State-of-the-art research on Eu^3+^-doped TiO_2_ nanoparticles predominantly employs wet-chemical methods like sol–gel, focusing on controlling europium concentration and phase composition to mitigate photoluminescence (PL) quenching. While these methods are often described as single-step, they typically involve reactions between titanium precursors (e.g., alkoxides) and europium precursors (e.g., chlorides or nitrates) with different reactivity and hydrolysis kinetics. This disparity can hinder the simultaneous formation of a uniform Eu-O-Ti matrix, often leading to the segregation of Eu-Eu clusters that reduce photoluminescence (PL). To address this issue, most syntheses use low europium concentrations, generally below 5 mol%. However, sol–gel methods excel in producing amorphous TiO_2_ matrices that offer an ideal environment for Eu^3+^ ions. Pioneering work by Frindell et al. showed that TiO_2_ doped with up to 8 mol% europium avoids quenching when structured into two adjacent phases: a crystalline phase, which acts as an antenna to sensitize Eu^3+^ emission, and an amorphous phase, where Eu^3+^ ions localize without quenching [6]. Yin et al. demonstrated that 5 mol% Eu-doped TiO_2_ mesoporous materials, prepared from [Ti(OtBu)_4_]_m_ and [Eu(OEt)_3_]_m_ and calcined at 400 °C, achieved outstanding PL efficiency due to a having semi-crystalline structure [7]. Pal et al. further explored the influence of europium concentration on PL in TiO_2_:Eu^3+^ nanophosphors synthesized by sol–gel, finding that PL intensity increased with Eu content up to 5 mol% but vanished after annealing at 500 °C, where crystallization into anatase nanocrystallites occurred [8]. Challenges remain then in achieving uniform Eu^3+^ dispersion and preventing Eu-Eu clustering due to precursor reactivity discrepancies.

Our research moves beyond the state of the art by leveraging heterometallic or single-source precursors (SSP) to achieve superior chemical homogeneity and uniform Eu^3+^ dispersion under mild synthesis conditions. This innovative approach not only prevents clustering but also facilitates optimized interfacial charge transfer, paving the way for novel functionalities in PL applications and beyond. The chemistry of alkoxides is well-known for producing heterometallic or single-source precursors (SSP) through Lewis acid-base reactions, promoting excellent chemical homogeneity under mild conditions. Our research previously demonstrated the synthesis of Gd- and Mg-doped TiO_2_ anatase nanocrystallites (5–10 nm) from heterometallic precursors, achieving high surface functionalization at low temperatures (100 °C). These nanomaterials showed promising results as MRI T1-contrast agents and as catalysts for nitrate hydrogenation [9,10].

In this study, we explored the sol–gel synthesis of europium-doped TiO_2_ nanoparticles using heterometallic Eu-Ti precursors, focusing on europium distribution at the surface and its effect on luminescent properties across varying europium concentrations. Compared to conventional surface impregnation, this method enabled a high Eu^3+^ ion loading (>10 mol%) without quenching photoluminescence, advancing the potential of Eu-doped TiO_2_ for optoelectronic applications.

## 2. Results

### 2.1. Eu-Ti Single-Source Precursor and As-Prepared Eu^3+^-Doped TiO_2_ Nanoparticles Characterizations

The mixed complex Eu_4_TiO(OiPr)_14_ was obtained by Lewis acid-base reaction mixing a stoichiometric amount of Eu_5_O(OiPr)_13_ and Ti(OiPr)_4_ in toluene at room temperature as previously described for the analog Sm_4_TiO(OiPr)_14_ [11]. Eu_4_TiO(OiPr)_14_ was characterized by FT-IR spectroscopy and elemental analysis (Eu, Ti). The FT-IR spectra of Eu_4_TiO(OiPr)_14_ show the appearance of new ν(M-O) bands between 500 and 700 cm^−1^ compared with the spectrum of Eu_5_O(OiPr)_14_ (see arrows in Appendix A), suggesting, by analogy with the results obtained with Sm_4_TiO(OiPr)_14_, the formation of a heterometallic compound. The Eu:Ti 4:1 stochiometry was also confirmed by elemental analysis.

A series of europium-loaded titania nanomaterials was synthesized by the sol–gel process from toluene solutions containing Ti(OiPr)_4_ and various equivalents of Eu_4_TiO(OiPr)_14_. After 20 h under stirring at room temperature, its addition to a refluxing aqueous solution of nBu_4_Br led to the immediate precipitation of white solids. These were collected by centrifugation after 5 min or 1 h of reflux and then washed with water and ethanol and dried at 70 °C for 12 h (Figure 1).

TiO_2_ obtained using the same sol–gel synthesis route but without the addition of Eu_4_TiO(OiPr)_14_ acted as a support for the impregnation of different quantities of Eu^3+^ in the presence of Eu(NO_3_)_3_.

The powder XRD patterns of all as-prepared (100 °C) sol–gel samples (Appendix A) displayed crystalline TiO_2_ phases; only the one which had been refluxed for 5 min (TEu10.0-0) was completely amorphous. It should be noted that most of the sol–gel syntheses published previously give amorphous Ln-doped TiO_2_ which then has to be calcined from 500 °C to provide the anatase phase [12].

Ln-Ti SSP strategy has already been used by Svensson et al. to obtain the uniform distribution of the dopant in the final Ln-doped TiO_2_ (Ln = Y, Sm, Eu) oxide, leading to more efficient photocatalysts than commercial P25 [13,14]. However, a calcination temperature in excess of 500 °C was required to obtain the anatase phase, which leads to much greater structural reorganization than in our study.

Thermogravimetric analysis in air up to 900 °C displayed an analogous behavior for all as-prepared samples with two major weight losses (20–200 °C and 200–400 °C), corresponding to adsorbed solvents and water and organic residues (-OH, -OR), respectively (Appendix A). It should be noted that the doped nanomaterials obtained by impregnation show much lower mass losses than those obtained by the sol–gel process, and that these increase with the amount of europium in direct relation to the proportion of amorphous material. It can therefore be suggested that the addition of europium affects the hydrolysis/polycondensation process and results in a greater presence of hydroxyl groups (-OH), confirmed by the significant loss of mass of the amorphous compound (TEu10.0-0). In order to compare the photoluminescence performances and to have a clean nanoparticle surface, the as-prepared powders were then finally calcined at 400 °C for 4 h under air.

### 2.2. Eu^3+^-Doped TiO_2_ Nanoparticles Characterizations

#### 2.2.1. Structural Properties

All the samples were first characterized by an elemental analysis of europium to determine the precise weight percent of dopant (X) (Appendix A).

The powder XRD patterns of the TEuX samples (X = 0.0, 0.5, 2.6, 5.2, 6.0, 9.1, and 11.0 mol%) (Figure 1) and TEuX-im samples (X = 2.7, 4.8, and 7.6 mol%) (Figure 2) after calcination for 4 h under air did not display significant modifications and showed that the materials mainly constituted TiO_2_ in the anatase phase along with some brookite content (Table 1).

The proportion of the brookite phase is around 30% for TEu0 and decreases with the addition of europium to become negligible or difficult to estimate from TEu5.2. Moreover, as the quantity of europium increased, the crystallites size for brookite decreased and the amorphous ratio or fluorescence effect increased. For all samples, no crystalline Eu_2_O_3_ peak was detected in the XRD patterns within the limit detection, even for the high doping level. Average crystallites estimations, according to the Scherrer equation, gave values between 6 and 9 nm for TiO_2_ anatase. Doping was then accompanied by a slight decrease in the crystallite size, which is in agreement with the literature, showing that a small amount of Ln^3+^ can inhibit TiO_2_ growth [16,17]. As our strategy allowed high-content doping, we could also detect that for Eu^3+^ concentrations in the range of 5.2–11.0 mol%, a difference in the line broadening of some [hkl] triplets could be interpreted by anisotropic size effects. It should be noted that only the TiO_2_ anatase parameter lattice shifted and this was only slightly, even at high doping percentages, suggesting that the europium atoms may not be located in the matrix, but mainly on the surface of the nanoparticles. In contrast, no change was visible for the powder XRD spectra of samples obtained by impregnation, with the proportion of brookite remaining stable until 7.6 mol% of Eu^3+^.

The SEM images of TEu0.5 and TEu11.0 show aggregates of spherical particles with fairly similar morphologies despite different europium contents (Appendix A). TEM images of the TEuX nanoparticles produced with different compositions of Eu^3+^ are shown in Figure 3. They indicated an average crystallite size of 6 nm, except for the TEu9.1 and TEu11.0 samples, which is close to the result of XRD. While the strong tendency for aggregation was similar for each TEuX sample, the increase in europium’s loading led to higher anisotropic morphology and higher particles size, as suggested by the Rietveld refinements. Such an effect of the Eu^3+^ doping onto the crystallographic orientation of TiO_2_ nanocrystals has already been attributed to the hindering of the growth of specific facets of anatase TiO_2_ based on the “oriented attachment” mechanism [17].

The STEM images of TEu9.1, TEu10.0-0, and TEu7.5-im samples, which are comparable in terms of their europium content, showed excellent Eu^3+^ dispersion at atomic scale (Figure 4). However, a detailed analysis of the images would suggest that while this dispersion is fairly homogeneous for the totally amorphous TEu10.0-0 sample (Figure 4C), it is slightly less so for samples that are partially crystallized, which could be explained by a greater localization in the amorphous parts of the TEu9.1 samples (Figure 4A,B).

Figure 5 compares the Eu3d_5/2_, Eu4d_5/2_, O1s, and C1s XPS spectra of samples TEu10.0-0, TEu9.1, and TEu4.8-im, the latter two having similar Eu^3+^ surface density (Table 1). For all of the spectra, the XPS features of Eu3d_5/2_ above 1134.3 eV and the shake-down satellite above 1125.7 eV were found at a higher binding energy (BE) than Eu_2_O_3_ (Figure 5A) (1133.7 and 1123.5 eV, respectively [18]). This increasing BE shift could be correlated to the vicinity of more electronegative Ti atoms in Eu-O-Ti units that pulled electrons away from Eu. For all the samples, quantitative analyses of surface composition based on Eu3d peaks or of volume composition based on Eu4d_5/2_ peaks differ greatly in terms of Eu^3+^ atomic percentages, demonstrating strong heterogeneity in the location of the dopant (Appendix A). Moreover, the europium atomic percentage measured by XPS (from Eu3d_5/2_ peaks) was between 14 and 16%, which was much higher than the one given by elemental analysis (around or less than 1 at %), suggesting a major contribution from the Eu atoms on the surface of the nanoparticles. The O1s XPS spectra were decomposed into three peaks at 529.7–529.9, 530.9–531.1, and 532.7–523.8 eV which could be assigned to the O^2−^, OH^−^, and oxygen vacancies [19] and the CO_3_^2−^ [20] oxygen species, respectively (Figure 5C and Appendix A). With regard to the other spectra, the TEu10-0 sample differs from the other two, with a higher amount of OH^−^ species. This result suggests that the amorphous TEu10.0-0 solid can be better described as an oxohydroxo species. The high resolution XPS C1s spectra were also resolved into three individual component bands at 284.9–285.0 eV (C-C), 286.1 eV (C-O), and 288.7–288.9 eV (COO), which indicated the contribution of carbonate ligands due to the europium atom at the surface of the nanoparticles (Figure 5D).

Figure 6 compares the Raman spectra of un-doped TiO_2_ and Eu-doped TiO_2_ samples prepared by the sol–gel method. For all the samples, except the amorphous one, the vibrational modes of the anatase TiO_2_ phase were mainly observed. They were located at 147 (Eg), 198 (Eg), 398 (B1g), 515 (A1g,B1g), and 637 (Eg) cm^−1^ [21,22]. The significant blue-shift of the main band compared to single crystal (147 cm^−1^ instead of 144 cm^−1^) is due to phonon confinement in TiO_2_ nanoparticles [23]. The typical bands of rutile TiO_2_ at 448, 612, and 827 cm^−1^ [22] were absent in all the spectra. However, small additional bands were evidenced near 244, 292, 319, and 361 cm^−1^. These bands arise from the brookite TiO_2_ phase [24], confirming the presence of this phase as revealed by XRD. They were less intense for the TEu5.2 sample and absent from TEu-9.1, suggesting that the addition of Eu^3+^ cations limits the formation of the brookite phase. For all the samples, the main bands of the crystalline Eu_2_O_3_ and Eu_2_Ti_2_O_7_ pyrochlore located, respectively, at 351 cm^−1^ [25,26] and 308 cm^−1^ [27], were not distinguished in particular in the spectra of the TEu11.0 sample for which the high Eu content could favor their formations. These features suggest again that Eu^3+^ cations were highly dispersed over TiO_2_ crystallites or incorporated into the TiO_2_ lattice. However, this last assumption can be ruled out, since no shift of the Raman bands was observed. Indeed, as confinement phonon, the presence of defects and non-stoichiometry can shift, broaden, and change the shape of Raman bands [28]. In particular, a blue-shift of 4 cm^−1^ was observed for the main Eg band of Eu-doped anatase TiO_2_ synthesized by induction thermal plasma pyrolysis [29]. In this case, the shift revealed Eu incorporation into the anatase lattice since the crystallite size remained high (35–60 nm), and, therefore, the confinement effect can be ruled out. A similar shift associated with broadening was reported by Pal et al. for TiO_2_:Eu^3+^ nanoparticles prepared by sol–gel method [8]. However, the observed phenomena could mainly arise from the decrease in nanocrystallite size and hence an increase in the confinement effect.

The Raman spectra of the Eu^3+^-doped materials obtained by impregnation coincide perfectly with those of the TiO_2_ support (Appendix A); they are composed of anatase and brookite, demonstrating that this doping method has no influence on the structure of the nanocrystallite, whatever the Eu(NO_3_)_3_ content.

#### 2.2.2. Textural Properties

The N_2_ adsorption–desorption isotherms, carried out on TEuX samples desorbed at 400 °C for 3 h, gave specific areas (SSA) ranging from 168 to 216 m^2^.g^−1^ (Table 1). These values were higher than for un-doped TiO_2_ (140 m^2^.g^−1^) in agreement with the observation reported by Sibu et al., suggesting the presence of lanthanide ions distributed uniformly onto the titania matrix [30]. The isotherms were all type IV which confirmed the presence of mesoporous structures attributed to intergranular porosity (Figure 7). However, the shape of the hysteresis evolves from an H2 type to an H1 one as the amount of Eu^3+^ increases, which can be attributed to a change in pore shape associated with anisotropic particle growth.

#### 2.2.3. Thermal Behaviors

After heating at 900 °C, the main phase of TEu2.6 (labeled TEu2.6-900) remained anatase along with rutile, brookite, and Eu_2_Ti_2_O_7_ traces, whereas un-doped TiO_2_ (TEu0-900) completely turned to rutile (Appendix A). Furthermore, the crystallite size increased to around 14–15 nm after heating at 900 °C due to the sintering phenomenon. It appeared that the prepared doped materials were thermally stable even at a high temperature, similarly to Zeng et al.’s observations for TiO_2_:Eu^3+^ nanocrystals [31]. The phase transformation of anatase-to-rutile was inhibited with the presence of Eu^3+^ because the existence of Eu-O-Ti bonds on the crystallite surface blocked the nucleation of rutile. Hence, Eu^3+^ doping was protected from sintering, which may be due to the Eu^3+^ ions being segregated in the grain boundaries of anatase TiO_2_ nanocrystals and to the increase in the diffusion barrier at the TiO_2_-TiO_2_ grain contact.

#### 2.2.4. Photoluminescence Properties

The luminescence mechanism of europium-doped TiO_2_ nanoparticles is governed by energy transfer from the TiO_2_ host to the europium ions, where the TiO_2_ absorbs UV light and generates electron–hole pairs. The energy is non-radiatively transferred to the Eu^3+^ ions, which then emit characteristic red luminescence via intra-4f transitions. As europium concentration increases, luminescence intensity initially rises due to the increased availability of dopant centers but eventually diminishes beyond an optimal concentration due to quenching effects like cross-relaxation and non-radiative energy transfer between closely spaced Eu^3+^ ions. This phenomenon is usually illustrated in an energy diagram referred to as a Jablonski diagram (Appendix A), showing the conduction band of TiO_2_ transferring energy to the excited states of Eu^3+^ followed by radiative de-excitation.

The evolution of the photoluminescence spectra with the Eu^3+^ content is plotted from 570 to 670 nm in Figure 8A,B. The bands observed for all the samples at 580, 592, 612, and 652 were attributed to ^5^D_0_→^7^F_0_, ^5^D_0_→^7^F_1_, ^5^D_0_→^7^F_2_, and ^5^D_0_→^7^F_3_ electronic transitions, respectively [8,17,32]. The 5D0→7F2 transition is very sensitive to structural changes and the environment of Eu^3+^ cations [33]. The corresponding band was very broad for all TEu samples (FWHM higher than 20 nm), which is typically found for Eu^3+^ cations in an oxide glass environment [34]. It suggests that Eu^3+^ cations were not located in the anatase crystalline lattice. Additionally, the ratio of the integrated intensities of the ^5^D_0_→^7^F_2_ (electric-dipole allowed) and ^5^D_0_→^7^F_1_ (magnetic-dipole allowed) transitions is sensitive to the asymmetry of the Eu^3+^ environment [33,35]. The high values (between 3.7 and 4.5) calculated for the TEuX samples were indicative of the low symmetry of Eu^3+^, confirming their location at the surface.

## 3. Discussion

Figure 9 shows the integrated intensity of the photoluminescence bands as a function of Eu^3+^ molar content. The literature frequently reports a decrease in intensity beyond a few percent Eu^3+^ due to a phenomenon called quenching, which arises when Eu^3+^ luminophores are close enough to facilitate cross-relaxation [31]. In this study, however, no quenching was observed up to a Eu molar concentration of 9.1–11%. This absence of quenching at such high Eu levels is rare: to our knowledge, it has only been reported in Eu^3+^-doped cubic mesoporous titania films [6,36] and TiO_2_:Eu^3+^ nanotubes [32]. The explanation lies in the preparation method used here, which promotes high Eu^3+^ dispersion across TiO_2_ nanocrystallites. As nearly all Eu^3+^ cations are surface-localized, an interesting parameter to note is the Eu surface density (Eu.nm^−2^) listed in Table 1, which increased only moderately with Eu molar concentration, was mitigated by the increase in BET surface area, helping to prevent quenching. This impact is evident for the TEu2.6-900 sample, where reduced photoluminescence intensity (Appendix A) is attributed to thermal treatment, which led to sintering and a drop in BET surface area, thus increasing Eu^3+^ surface density from 0.111 to 0.245 Eu.nm^−2^; this is consistent with previous findings by Ningthoujam et al. [27].

The data strongly indicate that Eu^3+^ atoms are predominantly localized in an amorphous environment near the TiO_2_ nanoparticle sub-surface. However, these factors alone do not fully explain the exceptional photoluminescence properties observed at high Eu^3+^ concentrations with this method, which successfully avoids quenching. As Figure 9 shows, merely impregnating high-surface area TiO_2_ with Eu^3+^ at concentrations above 4 mol% results in significantly lower PL intensities. Differences in optical performance, despite comparable surface densities (Table 1) and similar STEM images (Figure 4), confirm that our single-source precursor (SSP) strategy not only enhances atomic-scale Eu^3+^ dispersion but also optimally positions the Eu^3+^ emission centers at the TiO_2_ sub-surface. With both sol–gel precursors being alkoxides, it is likely that their hydrolysis rates are similar, which favors an optimized interfacial growth, unlike what is typically achieved with ionic europium salts. Another benefit of this approach is that it results in a crystalline core–amorphous shell structure at 100 °C, which is unusual. This configuration eliminates the need for high-temperature calcination, thereby avoiding the reorganization and segregation of Eu^3+^ ions that lead to quenching. Moreover, the significantly lower photoluminescence intensity of TEu10.0-0 compared to TEu9.1 emphasizes the importance of a crystalline TiO_2_ matrix at low temperatures for efficient Eu^3+^ emission. This nanocomposite structure enables more effective charge transfer than conventional impregnation methods, even with similar Eu^3+^ dispersion and surface density, which are typically quenching factors.

## 4. Materials and Methods

### 4.1. Synthesis

All the manipulations of air-sensitive derivatives were carried out under an inert atmosphere of argon using the Schlenk line technique. Ti(OiPr)_4_ was purchased from Sigma-Aldrich (Saint-Louis, MO, USA) (ref 87560, >97%) and distilled under vacuum before use. NBu_4_Br was used as purchased from Sigma-Aldrich (ref 86860, >99.0%). Eu chips were purchased from STREM and used without further purification. Eu_5_O(OiPr)_13_ was synthesized following the procedure detailed in reference [37]. Toluene was purified by a MBRAUN solvent purification system SPS-800 and isopropanol was distilled on aluminum isopropoxide ([Al(OiPr)_3_]_m_) and then stored over molecular sieves. Deionized water was used for hydrolysis.

#### 4.1.1. Synthesis of Eu_4_TiO(OiPr)_14_ (**1**)

A total of 1 mL (0.31 mmol) of Ti(OiPr)_4_ was mixed in 10 mL of toluene and 386 mg (0.25 mmol) of Eu_5_O(OiPr)_13_. The medium was stirred at room temperature for 20 h. The colorless solution was concentrated under vacuum (1/5). After being under −20 °C conditions for 48 h, 330 mg of colorless crystals of **1** (yield = 71%) was obtained. This compound, **1**, is highly soluble in hydrocarbons and alcohols and no single crystals could be obtained for X-ray structure determination.

Elemental analysis: Calcd for C_42_H_98_O_15_Eu_4_Ti (M.W. = 1497.51 g mol^−1^) Eu: 40.59, Ti: 3.18. Found Eu: 41.23, Ti: 3.25. FT-IR (Nujol) [cm^−1^]: 1333 sh, 1162 s, 1130 s (ν C-O), 998 s, 969 s, 911 m, 829 s; 771 w, 727 s; 616 w, 591 m, 524 m, 486 w, 464 w, 444 m (ν M-O).

#### 4.1.2. Synthesis of Eu^3+^-Doped TiO_2_ Nanoparticles

In typical synthesis, Eu_4_TiO(OiPr)_13_ was dissolved in toluene and different molar fractions of Ti(OiPr)_4_ were added. The homogeneous mixture was stirred for 20 h at room temperature. Then, it was quickly added to 100 mL of a NBu_4_Br (0.05 mol.L^−1^) refluxing aqueous solution under vigorous stirring: a white precipitate appeared immediately. The suspension was heated under reflux for 3 h and was then centrifuged to yield a white solid (Figure 1). These hydrolysis conditions in an aqueous medium with a high ionic strength have previously been optimized in our group to obtain crystallized TiO_2_ at 100 °C [38]. A sample, with 10 %mol Eu^3+^, was hydrolyzed at 100 °C for 5 min to check the importance of crystallinity on photoluminescence performance and was designated TEu10.0-0.

All as-prepared precipitates were washed by deionized water and then by ethanol, dried at 70 °C for 12 h, and finally annealed in a furnace at 400 °C under air in order to obtain reproducible and clean surfaces of the nanoparticles for the sake of comparison. The samples are denoted TEuX where X was the molar percentage of europium loading.

For comparison, several Eu^3+^ impregnated TiO_2_ nanomaterials were synthetized as follows: 1 g of TiO_2_ was stirred in 10 mL of a Eu(NO_3_)_3_ aqueous solution for 24 h and then the water was removed under vacuum. The solid was annealed at 400 °C under air for 4 h. These samples are labeled TEuX-im.

### 4.2. Characterization Methods

FT-IR spectra were recorded on a Bruker Vector 22 spectrometer. The precursors were emulsified in Nujol or Fluorolub between two KBr pellets and the powders from the hydrolysis were pelletized with KBr. TGA/TDA data were collected on a Setaram (Caluire, France) 92 system under air with a thermal ramp of 10 °C. min^−1^. The powder X-ray diffraction data of the TEuX samples were obtained on a Bruker (Billerica, MA, USA) D8 Advance A25 diffractometer equipped with a Ni filter (Cu Kα radiation: 0.154184 nm) and a one-dimensional multistrip detector (Lynxeye (Paris, France), 192 channels on 2.95°). Diffractograms were collected between 20 and 80° (2θ) with steps of 0.02047° and a total acquisition time of 110 min. Unit cell parameters of TiO_2_ phases, quantification, and crystallites size were obtained by using the Rietveld refinement method (FullProf Suite package [39]).

XPS experiments were performed with a KRATOS Axis Ultra spectrometer using monochromatic Kα-Al radiation as the excitation source. BE was calibrated by using the Ti2p_3/2_ level of TiO_2_ at 458.5 eV. TEM and SEM images were collected on a JEOL JEM-2010 transmission electron microscope and a JEOL JSM-5800LV scanning electron microscope, respectively. Scanning transmission electron microscope (STEM) images were acquired using a ThermoFisher (Waltham, MA, USA) TITAN ETEM G2 transmission electron microscope operating at 60–300 kV utilizing a dark-field detector with a camera length set to 245 mm. This configuration corresponded to collection angles in the range of 29 to 146 mrad. This microscope is equipped with an objective Cs aberration corrector, enhancing image resolution by reducing spherical aberrations. N_2_ adsorption–desorption BET experiments were carried out on an ASAP 2010 system (version 6.0) (Micromeritics) (Merignac, France) after desorbing the annealed samples at 400 °C for 3 h. Eu molar percentages were obtained from the “IRC@TECH platform” of IRCELYON by inductively coupled plasma optical emission spectroscopy (ICP-OES Activa from HORIBA Jobin Yvon) (Longjumeau, France) at 412.97, 381.697, and 393.048 nm wavelengths and CHN elementary analysis from “Service Central d’Analyses du CNRS” for air-sensitive precursors. Raman and photoluminescence spectra were achieved using a UV-Vis-NIR LabRam HR spectrometer (Horiba-Jobin Yvon) (Longjumeau, France) equipped with a BXFM confocal microscope, interference and Notch filters and a CCD detector cooled at –76 °C by the Peltier effect. The diffused light was spatially dispersed with a 1800 grooves.mm^−1^ diffraction grating for Raman and with 300 grooves.mm^−1^ for photoluminescence. The exciting line at 514.5 nm delivered by a 2018 RM Ar–Kr laser (Spectra physics) (Beaune-la-Rolande, France) was focused on powder samples with a ×100 microscope objective, allowing for the analysis of an area of ca 1 μm. Its power measured at the samples was of a 1 mW value for which the laser heating was negligible. For each sample, 5 different areas were characterized by Raman and photoluminescence spectroscopies to control their homogeneity at the micrometer scale.

## 5. Conclusions

In conclusion, synthesizing Eu^3+^-doped TiO_2_ nanoparticles from a Eu-Ti single-source precursor via the sol–gel process offers a highly efficient, single-step approach to creating luminophores with outstanding luminescence properties. The placement of the dopant and its surrounding environment are critical for maximizing energy transfer between TiO_2_ and the dopant. This straightforward, low-temperature strategy aligns well with efforts to enhance optical properties through innovative core–shell insulator/semiconductor heterostructure synthesis. It addresses the challenge of incorporating Ln^3+^ ions in semiconductor matrices like TiO_2_ while preventing detrimental Eu-Eu interactions that reduce PL intensity. These promising results open the door for potential applications in areas such as nanobioprobes, optoelectronics, and display technologies.

## Data Availability

All data are included in the mansucript or the Appendix A.

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
