# Peer review of "Enhanced Photoluminescence of Europium-Doped TiO2 Nanoparticles Using a Single-Source Precursor Strategy"

_molecules, 2024, doi:10.3390/molecules29245824_

Round 1

Reviewer 1 Report

Comments and Suggestions for Authors

This study provides a comprehensive investigation into the sol-gel synthesis of europium-doped TiO₂ nanoparticles, employing heterometallic Eu-Ti precursors. The authors focus on the distribution of europium at the surface and examine its impact on luminescent properties across various europium concentrations. Notably, their approach achieves a high Eu³⁺ ion loading (>10 mol%) without inducing photoluminescence quenching, a significant advancement compared to conventional surface impregnation techniques. This result enhances the potential of Eu-doped TiO₂ in optoelectronic applications by maintaining luminescent integrity at high dopant concentrations. The study's findings are supported by data indicating a high dispersion of Eu³⁺ ions within the TiO₂ nanocrystallites and their localization within an amorphous subsurface environment. Furthermore, the single-source precursor (SSP) strategy appears to effectively control atomic-scale Eu³⁺ distribution and optimally position emission centers, thereby promoting efficient energy transfer between the TiO₂ matrix and dopant. This study was conducted in a highly systematic manner, with all data of high quality. I suggest publication as soon as possible after addressing the following questions:

1. There is currently no evidence of the spatial distribution of the Eu element  from the morphology. Could the authors provide TEM-EDX mapping to illustrate the distribution of Eu within the nanoparticles?

2. What is the average size of the particles produced? It would be helpful if the authors could provide an average size estimation derived from XRD peak broadening.

3. A uniform doping profile can also help reduce the energy loss rate during thermalisation and relaxation processes upon excitation, thereby enhancing the radiative recombination efficiency of the system. It would be beneficial if the authors could consider citing this relevant paper: https://doi.org/10.1038/s41377-023-01236-w

4. The scale bar in Figure 3 could benefit from clarification, as it’s currently unclear whether the black or white bar represents 10 nm. Adjusting this for consistency would improve readability and interpretation of the image

Reviewer 2 Report

Comments and Suggestions for Authors

Violaine Mendez et al. investigated the photoluminescent characteristics of Europium-doped TiO2 nanoparticles. The samples were thoroughly characterized using Eu elemental analysis XRD, SEM, TEM, BET analysis, TGA, XPS, Raman spectroscopy, and photoluminescence measurements. Obtained results indicated enhanced efficiency of electro-optical properties attributed to the co-hydrolysis, which facilitates an optimized interface between the crystalline TiO₂ core and the dopant-rich amorphous surface, enabling far more effective charge transfer.

In my opinion, this manuscript can be published in your Journal after a minor revision.

1. In the Introduction part the authors must emphasize the state of the art and beyond state of the art,  to clearly delineate what is specific in their research.

2. The authors didn’t give any explanation about the BET results.

3. I suggest that SEM micrographs should be embedded in the manuscript.

4. Is there any possibility to do mapping?

5. In the photoluminescent part the authors should explain the concentration effect of the dopant, to give a deeper explanation of the mechanism of luminescence, energy diagram, etc…

Reviewer 3 Report

Comments and Suggestions for Authors

The ms titles ‘’ Enhanced Photoluminescence of Europium-Doped TiO2 nanoparticles via a Single Source Precursor Strategy’’ synthesis, structural and morphological characterization of TiO2 dope Eu3+ nanoparticles using a single-source precursor method, achieving enhanced photoluminescence through improved Eu³⁺ dispersion and minimized quenching. The results of this work suggest potential applications in optoelectronics due to the crystalline TiO₂ core and the dopant-rich amorphous surface structure and efficient charge transfer properties. The ms is well written and introduces a single-source precursor method for synthesizing Eu-doped TiO₂ nanoparticles, enhancing photoluminescent properties by facilitating high europium dispersion and preventing quenching at high concentrations. The methodology is thorough, including characterization techniques (XRD, TEM, SEM, XPS, BET, and photoluminescence analysis) and offering comprehensive data on particle morphology, crystallinity, and dopant distribution. The low-temperature synthesis and detailed comparative analysis between SSP and impregnation methods add depth to the study. I suggest that this ms should be considered for publication in Molecules after minor revision points that are listed below:

1)      The structure of the ms should be changed. The parts should be 1. Introduction, 2. Materials and Methods, 3. Results and discussion, 4. Conclusions. The results and discussion part could be divided into structural and morphological characterization and photoluminescence studies and all discussion of XRD, BET, Raman, etc. be transferred to the corresponding parts.

2)      Table 1 should be transferred to the part of surface and morphological characterization.

3)       Additional explanation regarding the interfacial charge transfer and structural implications of high Eu³⁺ dispersion would benefit the overall quality of the ms.

4)      Figures are generally well-prepared, but certain supplementary data, such as high-resolution TEM or SEM-EDX images or additional spectral comparisons, could provide clearer visual evidence of dopant distribution and structural integrity at higher Eu concentrations.

These adjustments will enhance the manuscript’s clarity, flow, and depth of analysis, making it well-suited for publication.

Reviewer 4 Report

Comments and Suggestions for Authors

Review of the manuscript Molecules-3324633-peer-review-v1: This manuscript investigates the improved photoluminescence of Europium doped Titanium Dioxide nanoparticles. The research is well focused, detailed, and offers a decent amount of novelty in this topic. Furthermore, language and style are fine. Some really minor things could be improved so my recommendation for now is minor revision.

Please see the following detailed comments:

Title – Satisfactory.

Abstract – The abstract is informative.

Introduction – Provides a clear and concise introduction to the topic. I would however add maybe just a few additional sentences/references as to why you chose Eu, and not another rare earth element.

Results and discussion

·       Debye Scherrer equation is just Scherer equation, please correct this.

·       “Doping was then accompanied by a slight decrease in the crystallite size which is in agreement with literature, showing that small amount of Ln3+ can inhibit the TiO2 growth [16].” Do you have a reference for Eu doping?

·       TEM images in Figure 3 seem to be quite out of focus, under or over focused?

·       Figures and Graphs: Unify the sizes and alignments of all graphs and TEM images for consistency. For example, Figures 4, 5, 9, and 12 vary significantly in size and placement.

·       Language: A minor language revision is recommended to improve readability and polish the text.

Materials and Methods – The section is detailed and adequately describes the experimental procedures.

Conclusions – The conclusions are well-written and effectively summarize the findings.

Literature – The literature cited is appropriate and relevant.

Author Response

We would like to thank the reviewers very warmly for the work done to improve the quality of the manuscript and for the very pertinent suggestions for improvement. We have tried to respond to the majority of requests within the time allowed for correction.

This manuscript investigates the improved photoluminescence of Europium doped Titanium Dioxide nanoparticles. The research is well focused, detailed, and offers a decent amount of novelty in this topic. Furthermore, language and style are fine. Some really minor things could be improved so my recommendation for now is minor revision.

Please see the following detailed comments:

Title – Satisfactory.

Abstract – The abstract is informative.

Introduction – Provides a clear and concise introduction to the topic. I would however add maybe just a few additional sentences/references as to why you chose Eu, and not another rare earth element.

Reply : Among the rare earths, europium is the most widely published for its luminescence properties, so there is a large body of publications and data. We therefore thought it would be interesting to test our preparation method using europium in order to benchmark it effectively and in a relevant way. From an applications point of view, europium-based nanophosphors can be used in lighting and medical imaging.

Results and discussion –

  • Debye Scherrer equation is just Scherer equation, please correct this.

Reply : this has been amended

  • “Doping was then accompanied by a slight decrease in the crystallite size which is in agreement with literature, showing that small amount of Ln3+ can inhibit the TiO2 growth [16].” Do you have a reference for Eu doping?

Reply : reference 17 has been added

  • TEM images in Figure 3 seem to be quite out of focus, under or over focused?

Reply: The images were enlarged to the maximum possible resolution of the microscope in order to check their morphology.

  • Figures and Graphs: Unify the sizes and alignments of all graphs and TEM images for consistency. For example, Figures 4, 5, 9, and 12 vary significantly in size and placement.

Reply: All the figures have been aligned and their sizes may vary when paginated.

  • Language: A minor language revision is recommended to improve readability and polish the text.

Reply: The text has been proofread and corrected as far as we are able to do so.

Materials and Methods 

The section is detailed and adequately describes the experimental procedures.

Conclusions – The conclusions are well-written and effectively summarize the findings.

Literature – The literature cited is appropriate and relevant.